# Documentation: A Security Tool for the Identification and Repatriation of Illicitly Trafficked Objects from Museums with Particular Reference to the National Gallery of Zimbabwe

**Davison Chiwara**

Midlands State University, Department of Archaeology, Cultural Heritage and Museum Studies, Senga Road, Gweru P.Bag 9055, Zimbabwe; davisonchiwara87@gmail.com

**Abstract:** The aim of the study was to ascertain how documentation assisted in the recovery of stolen and illicitly trafficked objects at the National Gallery of Zimbabwe. The research objectives were to: Assess the adequacy of the gallery's documentation system; appraise the documentation elements that were key in the tracking and repatriation of stolen objects to the gallery; analyze the gallery's networks on the documentation and safeguarding of objects against theft; and ascertain challenges faced by the gallery in documenting its objects. Research results indicated that the gallery's documentation system encompassed both manual and electronic documentation, which addressed vital aspects that have been prescribed by Object ID Standards for the identification and recovery of stolen objects. These include photographs, provenance, and name of objects. This, coupled with networks with key institutions involved in the fight against theft and illicit trafficking of objects, has enabled the gallery to recover its stolen objects from Poland. However, lack of state-of-the-art cameras has led to the production of poor photographs, which compromised the gallery's claim to its stolen objects. Additionally, lack of ideal software for the gallery's database is hampering effective documentation efforts. The research recommends that the gallery should acquire appropriate cameras for quality documentation of its objects and purchase database software with backup support for software upgrades to prevent loss of information on its objects.

**Keywords:** documentation; illicit trafficking; collections security

---

## 1. Introduction

Theft and illicit trafficking of objects from museums is widespread, occurring across the globe. This problem is not a recent phenomenon, as it has been noted in the past. In a preliminary study to determine the scope of the problem posed by the theft of and illicit traffic in cultural property, to which twenty-six countries across the globe responded [1], reported that thefts in museums that are mostly private have been widespread and countries such as Belgium, Finland, France, and Luxembourg have been affected. Further, according to a research done by the House of Commons in 2000, the illicit trafficking of antiquities is estimated to be above US$6 billion per year [2]. Zimbabwe is not immune to this problem, as objects have been stolen from national museums, some of which have been illicitly trafficked and have never been accounted for. Some of the stolen objects include the One Thousand Guinea Trophy containing 175 ounces of gold, King Lobengula's gold bracelet, Missionary Robert Moffat's gold watch, seven leopard skins (stolen from the Natural History Museum in Bulawayo), six gold crucibles (stolen from the National Mining Museum in Kwekwe), a gold bracelet (stolen from the Great Zimbabwe Museum in Masvingo), and a copper cross (stolen from the Zimbabwe Museum

of Human Sciences in Harare) [3]. A forensic audit conducted in 2006 revealed that nearly 1500 objects have been stolen from the Zimbabwe Museum of Human Sciences [4].

Illicit trafficking is a very complex problem, which must be dealt with from many sides. One of the means of fighting illicit trafficking is adequate and properly secured inventories [5]. Particular focus on the combat of illicit trafficking of cultural property should be on its prevention. This includes the creation of up-to-date inventories of national cultural heritage. Fighting against illicit trafficking is a process which starts even before an object is stolen or illicitly exported, as today's preventive measures will be tomorrow's evidence to support a claim [6]. There is also need for regular educational campaigns to stimulate and develop respect for cultural heritage and raise awareness of laws and issues relating to illicit trafficking [7]. Thus, inventories are a vital component of a documentation system that help in the recovery of illicitly trafficked objects. Without adequate documentation, it is it difficult for museums to claim back their stolen objects. INTERPOL (1974) [1] reported that in the United Kingdom, the discovery of stolen objects depended mainly on the documentation details provided by the owners and between 1968–1973, stolen works of art worth a total 12 million pounds were recovered. In this regard, accurate and accessible documentation helps in the tracking and identification of stolen museum objects by their rightful owners. Without this documentation, objects are at risk of theft and illicit trafficking in museums, and it is impossible to recover them.

Documentation is key to the protection of cultural objects, as stolen objects that have not been photographed and adequately described are rarely recovered by their rightful owners [8]. Accurate and accessible documentation is an essential resource for collections management, research, and public service [9]. Against this background, the research had the following objectives:

- To assess the adequacy of the gallery's documentation system;
- To appraise the documentation categories that were key in the tracking and repatriation of stolen objects to the gallery;
- To analyze the gallery's networks on the documentation and safeguarding of objects against theft and illicit trafficking;
- To ascertain challenges faced by the gallery in documenting its objects.

## 2. Documentation of Objects and Its Importance in the Fight Against Illicit Trafficking

The registration of a relevant cultural property in an inventory is a useful measure in preventing its trafficking and achieving the recovery of trafficked objects. Registration facilitates the identification of cultural objects by the authorities, supports the police and other control agencies in monitoring cultural objects and determining provenance, and provides a sound basis for claims for restitution and return [10].

Inventories must have adequate information to allow verifiable identification of an object when found by public authorities as suspect or offered for sale either locally or abroad. Of significance to note in the documentation of the objects is the inclusion of an authentication of their origin and a photograph, digital image or a drawing of the objects [5,9]. These have been agreed upon by international museums, police, customs, insurance, and data base experts and are included in Object ID [5]. Object ID is a core standard created for the very specific purpose of describing cultural objects for identification purposes [8]. Object ID is an effective instrument to describe archaeological, artistic, and cultural objects in order to facilitate their identification in case of theft. It also allows INTERPOL to have a more effective update of its stolen objects database if the object that disappeared was catalogued using the proposed guidelines [11].

On the other hand, the accessioning process is a key stage in the overall documentation of objects, which records the legal evidence for the ownership of the items in the collection and provides the starting point for the fuller cataloguing of individual objects. Objects should be recorded in accession files and an accession register capturing the accession number, date, source, method of accessioning, brief description of objects, and the name or initials of the museum curator [9]. Records for

specific objects need to be designed into distinct categories which hold a specific piece of information. The main subject areas in museums that are documented are archaeology, antiquities, ethnology, fine and decorative art, costume, history, and natural history collections. Regardless of the subject area, all records should include Object Number, Object Name, Current Location, and Distinguishing Features. These are vital for security purposes in cases where the object is stolen or misplaced. Other equally important fields are Title field for Art collections, the Production Period/Date field for archaeology collections, and the Classified Name field for natural history collections [9].

Documenting an object enriches its intrinsic value. The recording of information gives an object meaning and context and results in a stronger understanding of its uniqueness, its contribution to the collection, and its reason for being collected in a museum. Apart from that, documentation helps in the determination of the legal ownership of an object. During acquisition, the owner or donor of an object signs a document transferring legal title to the museum. This transfer of ownership is recorded in the museum register and the museum catalogue [12].

Elsewhere, dealers in cultural objects need to keep registers with the provenance and previous ownership of any object in their possession, including names and addresses of purchasers [5]. This documentation is vital, as it enables them to trace the history of the objects offered to them for sale as well as identify objects with a questionable history and alert law enforcement authorities to assess whether such objects were stolen and illicitly trafficked.

The documentation of the location of objects aids in their security and retrieval. The documentation of an object's location acts as a tracking device, mapping the movement of an item during its life in the collection. In the event of theft, the worth of documentation, particularly photographs, is immense. A precise list of objects can be presented to insurers and police. Detailed collection information helps insurers to value the museum's loss, and also helps the police in investigating the theft. This eventually helps in identifying and returning stolen items [12].

## 3. Methodology

Triangulation was used in data collection. This involved the use of two research instruments, namely questionnaires and interviews, to gather information. Both questionnaires and interviews were structured. Questionnaires were distributed to 8 curatorial, exhibitions, and conservation staff members. All of the questionnaires were completed and returned. These were augmented by 3 interviews conducted with heads of departments at the gallery, namely the Curatorship, Collections and Conservation, and Workshop and Maintenance Departments. Both questionnaires and interviews were used as research instruments because they ensured the validation of data provided by the research participants through cross-verification of these two research instruments. They also ensured a comprehensive understanding of documentation and the fight against theft and illicit trafficking at the gallery.

## 4. Research Findings and Discussion

### 4.1. The Adequacy of the Gallery's Documentation System

Nine research participants highlighted that the documentation system of the gallery consists of both manual and electronic documentation, which includes accession register, accession cards, accession number on objects, and a computerized database. These information sources on objects are interdependent and they enable cross reference searches to be carried out [13]. The categories of information covered by the gallery's documentation system derive from Object ID Standards on documentation. These include: Title of the object; maker of the object/artist; type of object (e.g., sculpture or painting); dimensions of the object (height, length, width); markings (signature or numbers), distinguishing features; origin; condition of the object; acquisition date; accessioning number; date of manufacture and period of use; material used in making the object; and location of the object within the gallery and photographs. In this regard, the information categories addressed by the

gallery's documentation system are comprehensive and capture a lot of details about an object. In cases where objects have been stolen and there is a need to lay claims, the gallery has many reference points for the identification and recovery of its objects. Thus, its documentation system is very adequate for identifying and claiming objects stolen from the gallery, as was the case in the recovery of the four headrests and two Makonde masks that were stolen and illegally trafficked to Poland (See Figures 1–4).

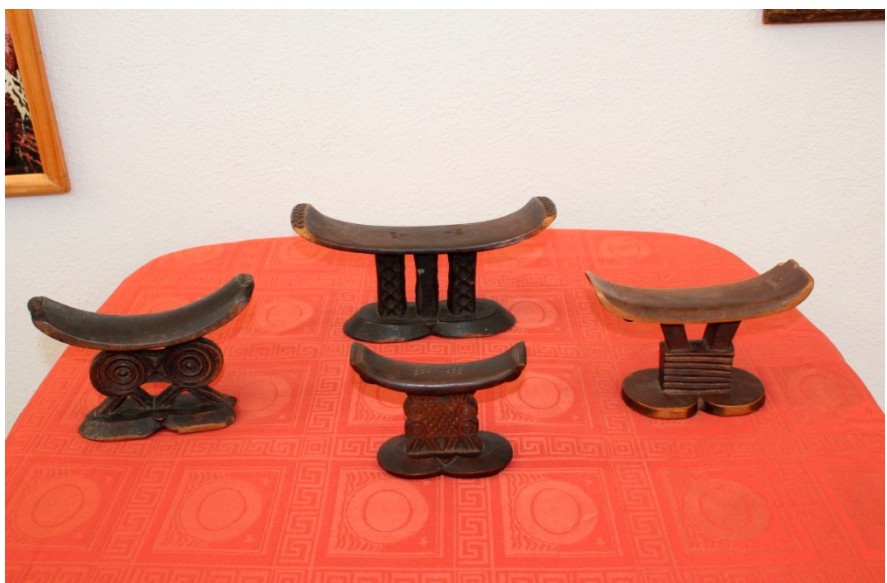

**Figure 1.** Stolen Zimbabwean headrests that were repatriated from Poland back to the gallery. Source: National Gallery of Zimbabwe.

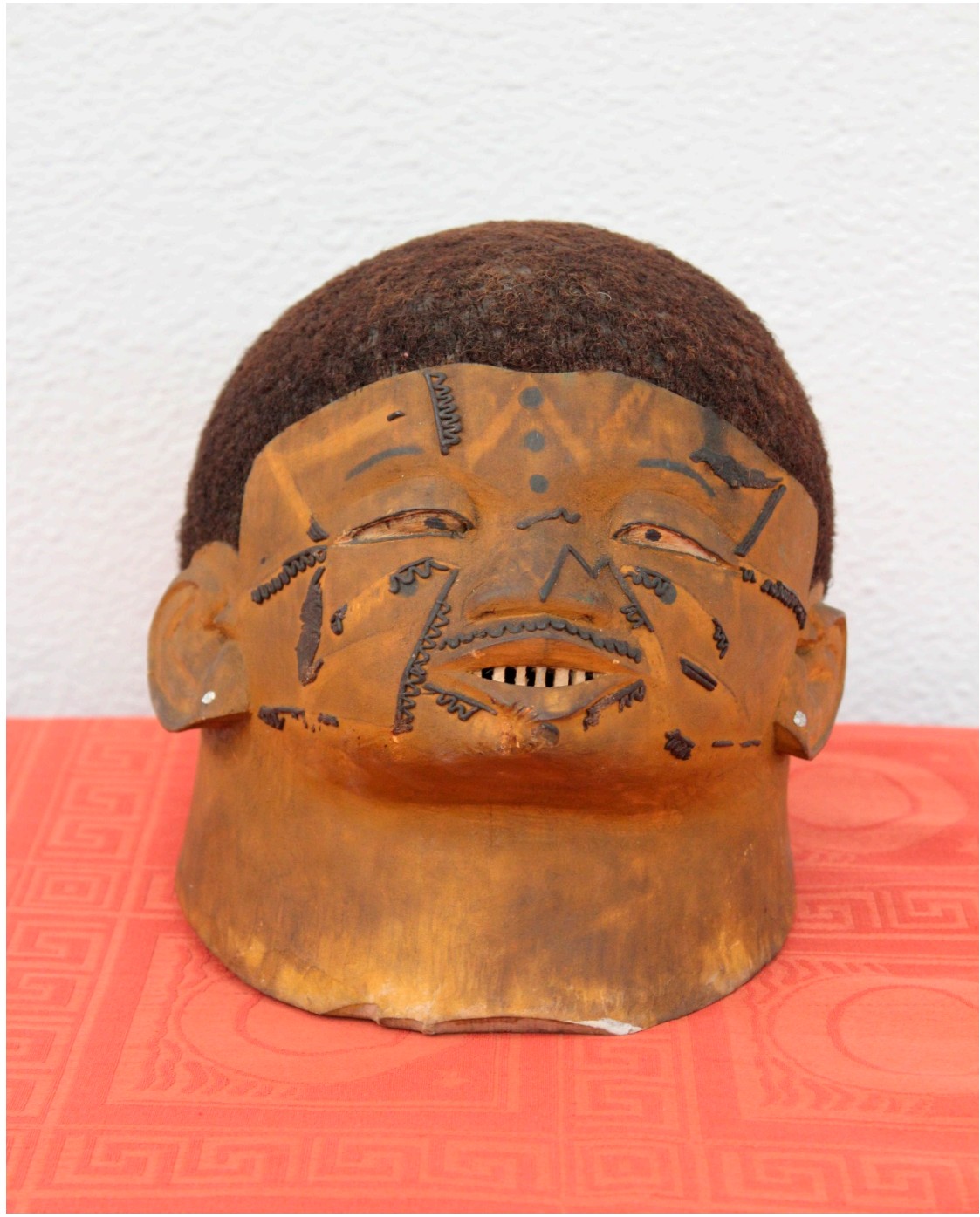

**Figure 2.** Stolen Makonde mask that was repatriated from Poland back to the gallery. Source: National Gallery of Zimbabwe.

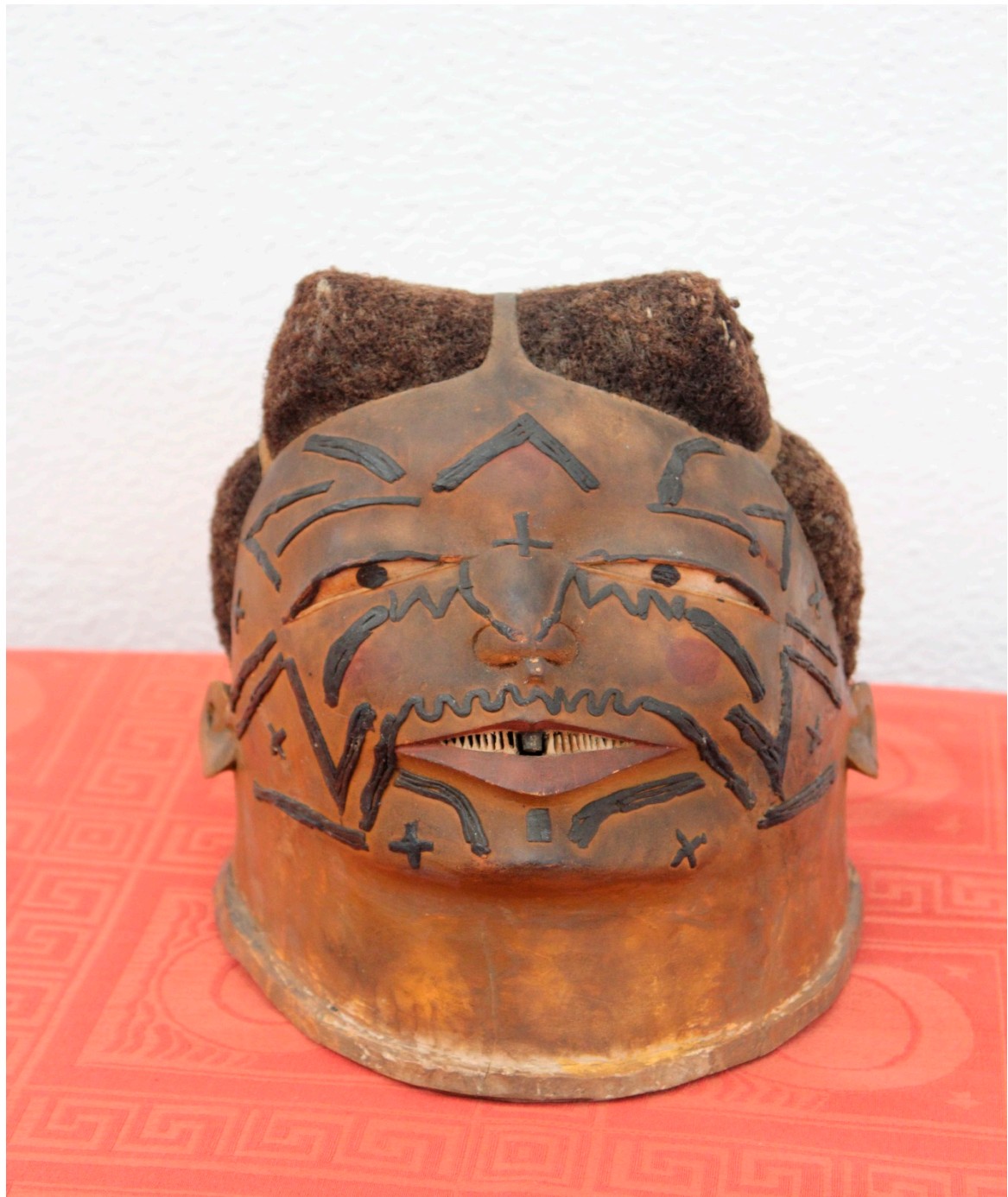

**Figure 3.** Stolen Makonde mask that was repatriated from Poland back to the gallery. Source: National Gallery of Zimbabwe.

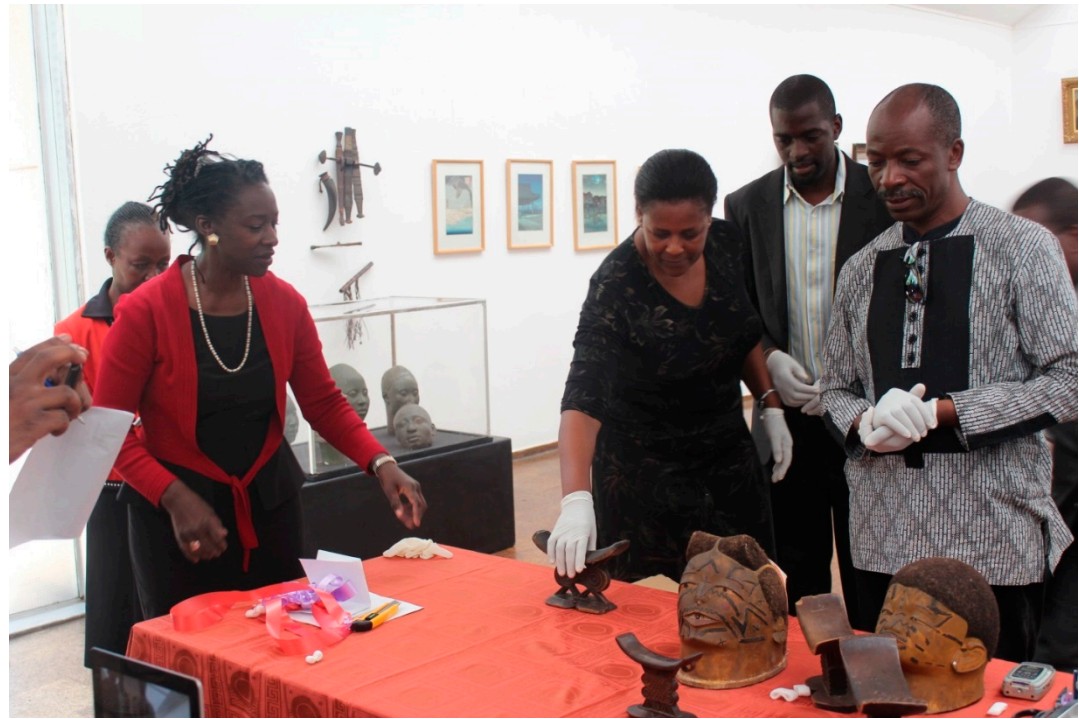

**Figure 4.** National Gallery of Zimbabwe, Harare staff receiving objects repatriated from Poland. Source: National Gallery of Zimbabwe.

*4.2. Key Documentation Categories and Synchronization of National and International Databases in the Tracking and Recovery of Stolen Objects*

Most of the research participants singled out photographs and object accession numbers as vital documentation categories that helped them in the claim and recovery of the objects stolen from the gallery. The participants indicated that in 2006, the gallery lost six ethnographic objects, which were four Zimbabwean headrests and two Makonde masks. They indicated that the stolen objects were later recovered in 2007 but were repatriated back to the gallery in 2013. See Figures 1–4.

The head of the Collection and Conservation Department at the gallery indicated that photographs were vital in the gallery's claim of the stolen objects. The departmental head explained that when the theft of the objects occurred in 2006, the gallery's documentation did not include photographs. She had taken personal photographs of the collection that was on display. It is these photographs that helped in the identification and recovery of the stolen objects from the gallery.

Photographic images, digital images, and scientific drawings of objects are vital documentation resources which are useful to customs officials and the media if an object is stolen [6,9]. These documentation elements are key and without them, it is difficult for museums to claim back stolen objects. A case in point is where the National Museums and Monuments of Zimbabwe (NMMZ) in 2005 failed in its endeavor to claim back ethnographic objects that were stolen from the Zimbabwe Museum of Human Sciences. The objects were located at an art market in Cape Town, South Africa. However, due to inadequate documentation, the NMMZ failed to get the dealer found with these objects convicted of theft of cultural property. NMMZ also failed to claim back and repatriate the stolen objects back to Zimbabwe, as it did not produce enough documentation evidence to South Africa Authorities [3].

Other participants also indicated that accession numbers on stolen objects helped the gallery in its claims of ownership. They indicated that all objects in the permanent collection have a unique code known as the permanent collection code that identifies them. This is a code that is assigned to an object when it is accessioned by the gallery. This code is affixed to the object in a hidden position known by the gallery. The participants indicated that they availed these permanent collection codes to

INTERPOL and law enforcement authorities in Poland. It was these codes that convinced them that the objects were owned by the gallery.

Further, these key documentation categories need to be shared in national and international databases to enable museums to easily trace collections in cases of theft. States should consider establishing unified national inventories, which should be comprehensive and coordinated and should include all relevant publicly owned cultural properties and private collections and properties. Such inventories could consolidate existing registers and catalogues and connect existing inventories, both public and private, in a coordinated network of national databases. These could be made accessible to all national and foreign public authorities through a multilingual online portal of a digital database to facilitate easy access and verification. As a reference point, states could use international standards, such as Object ID, which describes cultural objects and includes photographs of the works of art in question, a brief description of it, and a classification of the type of the object and the materials and techniques used to create it. These databases should also be linked with international databases, especially the INTERPOL Stolen Works of Art Database [10]. The establishment of these inventories at a national level would help prosecutors and judges to have a clear determination of ownership, and therefore of evidence of theft, by having inventories endorsed by multiple States, which could lead to more efficient prosecutions of trafficked, illicitly exported or imported, stolen, looted or illicitly excavated and illicitly traded cultural property [7,10,14]. Additionally, art and antiquities dealers as well as museums and other professionals could consult these databases before engaging in any professional dealings regarding any cultural property. This would greatly reduce the risk of good faith purchasers of stolen goods [10,15].

### 4.3. The Gallery's Networks on the Documentation and Safeguarding of Objects against Theft and Illicit Trafficking

Ten of the research participants explained that the gallery has a sound working relationship with organizations such the Zimbabwe Republic Police, The Museum Security Network in the Netherlands, and INTERPOL, with which it shared its documentation of the stolen objects. Documentation of the gallery's stolen objects was entered in INTERPOL's database. Databases such as Interpol's Stolen Works of Art Database are useful in helping to identify and return stolen property [15]. These networks helped in publicizing and raising alarm on the theft of the objects from the gallery. The Museum Security Network in the Netherlands posted the photographs of the stolen objects on its website to raise international alarm about the theft. The participants highlighted that coordinated efforts involving the gallery, the Zimbabwe Republic Police, the Museum Security Network, INTERPOL, and law enforcement authorities in Poland resulted in the discovery of the stolen objects in Poland as well as the arrest and subsequent conviction of the thief. This coordination shows that international coherence is needed between national legislation and policies in order to counter international crime carried out across multiple jurisdictions [15].

All of the research participants indicated that the gallery also has a cordial relationship with UNESCO, the National Museums and Monuments of Zimbabwe, and the National Archives of Zimbabwe. Five research participants highlighted that UNESCO conducts workshops involving the gallery, National Museums and Monuments of Zimbabwe, and the National Archives of Zimbabwe. They indicated that such workshops emphasize the need for accurate and up-to-date documentation of objects as well as local and international cooperation in the tracking and recovery of stolen cultural property.

### 4.4. Challenges Faced by the Gallery in Documenting Its Objects

One respondent pointed out that lack of state-of-the-art cameras led to the production of photographs of objects that were not very clear, which compromised the gallery's claim of the objects from Poland. During the time when its objects were stolen, the gallery relied on personal photographs of the head of Collection and Conservation Department to claim the stolen objects. Therefore, there is

a need for the gallery to acquire state-of-the-art cameras that produce quality photographs of objects, which makes it easy for the gallery to claim its objects if they are stolen. The gallery could try to look up philanthropic organizations to assist in the technological protection of its artworks. The challenge in Zimbabwe is that such organizations which are needed to help museums are rarely found. However, there are foreign embassies that have been actively involved in the funding of management of heritage in Zimbabwe, such as the United States of America Embassy in Zimbabwe. Engaging such embassies may help the gallery in securing funding for equipment for the documentation of its artworks.

The other challenge raised was that of lack of ideal electronic database software for documenting the gallery's objects. Most participants highlighted that the gallery uses File Maker Pro software for its electronic database. One participant explained that although this is not the ideal database software that the gallery needs, it has enabled the gallery to capture essential information about the objects, such as their origin, medium, and material makeup, which makes it easy to trace their provenance. However, all the participants pointed out that File Maker Pro software does not have image support features; thus, there is no photographic documentation of objects. The participants explained that the gallery now has a provision for separate images for its objects, which complements the electronic database documentation. Further participants explained that efforts have been made by the gallery to acquire more advanced software for its database, such as Museum Plus. The challenge is that of lack of backup support in Africa; thus, the gallery is left with no option but to use File Maker Pro software. This software does not have the provision for backup support from its manufacturer. In some instances, electronic databases can crash, and without backup support, the documentation for objects will be lost, which presents challenges in claiming stolen objects. When purchasing database software, there is a need to check whether a backup service is available to provide assistance and upgrades, and whether this is free [12].

Different software offers varying levels of sophistication and is priced to match that sophistication. Some software provides features such as efficient searching and reporting, image management, and simple image-linking to records [12]. File Maker Pro used by the National Gallery of Zimbabwe does not have features such as image management and image-linking to records. There is a need for the gallery to acquire software with these features to ensure that electronic documentation of its objects is complete. Photographic documentation is vital in tracking stolen objects as well as claiming them back from illegal owners. UNESCO (n.d) [6] highlights that in developing countries, attempts to document collections electronically fail when there is no long-term strategy for technical maintenance and adequate human resources. In this regard, the gallery needs to address these issues to ensure that its electronic documentation is secure.

## 5. Conclusions

The National Gallery of Zimbabwe in Harare has an adequate documentation system. Its documentation system comprises both manual and electronic documentation. The manual documentation system consists of object accession numbers, an accession register, and card catalogues for individual objects. Both the manual and electronic documentation complement each other and are used for cross referencing. Thus, if any element of its documentation system is compromised, the gallery has enough backup documentation. Further, the documentation categories of the gallery's documentation system derive from Object ID standards and they are very extensive, which ensures that the gallery is in a good position to identify and claim its objects in cases of theft and illicit trafficking.

The key documentation categories that helped the gallery in the identification and claim of its stolen objects that were recovered in Poland were photographs and subtle accession numbers affixed to objects. These key documentation categories helped authorities in the search and eventual recovery of the objects.

Documentation coupled with networking with other organizations helped in the tracking and recovery of the objects that were stolen from the gallery. These networks include the Zimbabwe Republic Police, The Museum Security Network of Netherlands, and INTERPOL. Other network

partners that work with the gallery on issues of documentation and security of objects include UNESCO, the National Museums and Monuments of Zimbabwe, and the National Archives of Zimbabwe.

Although the National Gallery of Zimbabwe has a sound documentation system, the lack of state-of-the-art cameras for documenting its objects compromises the quality of photographs that are taken. This may affect the gallery's claims on its stolen objects. Additionally, the lack of a backup of its electronic database hampers effective documentation of the objects. Databases that lack software backup may crash, resulting in the loss of vital information on objects.

**Funding:** This research received no external funding.

**Acknowledgments:** I would like to acknowledge the National Gallery of Zimbabwe, Harare staff for the support they gave me in this research.

**Conflicts of Interest:** The author declares no conflict of interest.

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
