# Peer review of "Documentation: A Security Tool for the Identification and Repatriation of Illicitly Trafficked Objects from Museums with Particular Reference to the National Gallery of Zimbabwe"

_heritage, doi:10.3390/heritage2010027_

Round 1
Reviewer 1 Report
Perhaps you can add more regional context to your paper - apart from the Zimbabwean case studies (NMMZ & NGZ) of stolen cultural property where else in the continent do we have example of stolen and recovered objects?
Your methodology need a bit of more elaboration - how did you conduct the interviews - were the interviews structured or they were just dialogical conversation with gallery workers? I don't know how effective the questionnaire method is, again there is need to give more detail on how they were administered and how the data was analysed.
Can you also provide photo credits to all the pictures used in text and permission of use by the NGZ?
Lack of support to buy the necessary softwares - whose prerogative is it? Does the gallery request the money from the government or it relies on donor funding (embassies) to support its technical activities?
What role does physical security details play in the whole documentation discourse at NGZ and NMMZ? Are they adequately trained in documentation standards and do they know the difference between an object in collection and a non - collection? Can give you some reflections on security details who guard these premises - is there a possibility that some of them might be accomplices?
Recently, NMMZ installed an alarm system outsourced from a private security company and yet it has its own security details who are suppose to provide security to the collections - what would you think are the implications of this move in relationship to your argument?
And apart from the need to install hight tech camera does the NGZ has security details who are well informed about the documentation system?
Author Response
Dear reviewer
Thank you for the constructive insights. I have managed to make some corrections on my my methodology as you highlighted. I have also included the photo credits that you highlighted. However, i could not manage to take a regional focus on my paper as it was largely focusing on Zimbabwe. I believe such a research needs more time and resources and i am willing to broaden my research to include the Southern African Region in the future.
Reviewer 2 Report
It is very interesting to know that museums in Zimbabwe are improving their documentation system in order to prevent the theft of items. As the author says, this is the best way to prevent the illicit traffic of cultural goods.
Author Response
Dear Reviewer
Thank you for your review.
Reviewer 3 Report
The value, process and conclusions of this study are all very clearly explained. It is a "basic" study with "unsurprising" conclusions, but those facts do not alter the necessity of the study or the value of its findings.
It also provides very clear examples of success and failure in the protection and recovery of cultural objects. These serve as lessons for other places, which have not yet achieved the same level of preparedness as Zimbabwe.
There need to be a few tiny changes in language. For example, "since 1968-1973 stolen works of art worth a total 12 million pounds have been recovered" sounds like that is the current total; but obviously, as the Interpol (1974) citation shows, those artworks were recovered between 1968 and 1974. Prot (2004) should be Prott (2004). Such changes are nothing more than proofreading.
The text feels a bit repetitive, but that may be an inevitable product of the nature of the study
Could a (blank) copy of the questionnaire be included as supplementary material, to make it easier for other professionals/institutions to conduct similar studies elsewhere?
Author Response
Dear Reviewer
Thank you for the constructive insights. I have done some corrections which you highlighted in your review. These include the spelling, language and methodology. I have also included the blank questionnaire that you asked for. Please find it here:
Documentation: A Security Tool for the Identification and Repatriation of Illicitly Trafficked Objects from Museums with Particular Reference to the National Gallery of Zimbabwe
Questionnaire
1. What is your documentation system composed of?
2. Is your documentation system adequate in the identification and recovery of objects illicitly trafficked from the gallery?
3. Which documentation elements were key in the tracking and recovery of objects stolen from the gallery?
4. Who are your networks in the documentation and safeguarding of objects against theft and illicit trafficking?
5. Are there any challenges that are faced by the gallery in the documentation of its objects?
Reviewer 4 Report
This is a useful and important paper. It fills a clear knowledge gap and is of international significance. If published, I will place this article on the reading list for my topic 'The Museum'. The argument could be strengthened slightly by drawing on the responsibility/capacity of philanthropic organisations to assist the museum in improving its technological protections, at a reasonably low cost.
The English is excellent, overall. The could be minor polishing changes, such as removing the word 'got' from line 261, but a final read would pick up these small errors of expression.
Author Response
Dear Reviewer
I hope you are fine. Thank you for the constructive insights in your review. I have removed the word "got" that you highlighted. I have also included the argument on philanthropic organisations in my manuscript.